# *Momordica charantia* fruit reduces plasma fructosamine whereas stems and leaves increase plasma insulin in adult mildly diabetic obese Göttingen Minipigs

Sietse Jan Koopmans[1]*, Gisabeth Binnendijk[1], Allison Ledoux[2], Young Hae Choi[2,3], Jurriaan J. Mes[4], Xiaonan Guan[5], Francesc Molist[5], Tâm Phạm Thị Minh[6], Nikkie van der Wielen[7]

1 Wageningen Livestock Research, Wageningen University & Research, Wageningen, The Netherlands, 2 Natural Products Laboratory, Institute of Biology, Leiden University, Leiden, The Netherlands, 3 College of Pharmacy, Kyung Hee University, Seoul, Republic of Korea, 4 Wageningen Food & Biobased Research, Wageningen University & Research, Wageningen, The Netherlands, 5 Schothorst Feed Research, Lelystad, The Netherlands, 6 Department of Food crops and Horticulture, Nong Lam University, Ho Chi Minh City, Vietnam, 7 Department of Animal Nutrition and Division of Human Nutrition, Wageningen University & Research, Wageningen, The Netherlands

* sietsejan.koopmans@wur.nl

**Data Availability Statement:** Individual pig data may be found in excel S1 File.xlsx.

## Abstract

### Background

Traditionally *Momordica charantia* (Bitter gourd) is known for its blood glucose lowering potential. This has been validated by many previous studies based on rodent models but human trials are less convincing and the physiological mechanisms underlying the bioactivity of Bitter gourd are still unclear. The present study compared the effects of whole fruit or stems-leaves from five different Bitter gourd cultivars on metabolic control in adult diabetic obese Göttingen Minipigs.

### Methods

Twenty streptozotocin-induced diabetic (D) obese Minipigs (body weight ~85 kg) were sub-divided in mildly and overtly D pigs and fed 500 g of obesogenic diet per day for a period of three weeks, supplemented with 20 g dried powdered Bitter gourd or 20 g dried powdered grass as isoenergetic control in a cross-over, within-subject design.

### Results

Bitter gourd fruit from the cultivars "Palee" and "Good healthy" reduced plasma fructosamine concentrations in all pigs combined (from 450±48 to 423±53 and 490±50 to 404±48 μmol/L, both p<0.03, respectively) indicating improved glycemic control by 6% and 17%. These effects were statistically confirmed in mildly D pigs but not in overtly D pigs. In mildly D pigs, the other three cultivars of fruit showed consistent numerical but no significant improvements in glycemic control. The composition of Bitter gourd fruit was studied by

**Funding:** This research project "Green Health Solutions" was financially supported by the European Regional Development Fund of the European Union (KVW-00117), a contribution of the Province Zuid-Holland and municipality Almere and by the KB-34 program "Circular and Climate Neutral society" of Wageningen University and Research (KB-34-009-003). There was no role of the funding bodies in the design of the study and collection, analysis, and interpretation of data and in writing the manuscript.

**Competing interests:** The authors have declared that no competing interests exist.

**Abbreviations:** A.I., arbitrary units; AUC, area under the curve; D, diabetic; GE, gross energy; SGLT2, glucose transport protein; STZ, streptozotocin.

metabolomics profiling and analysis identified three metabolites from the class of triterpenoids (Xuedanoside H, Acutoside A, Karaviloside IX) that were increased in the cultivars "Palee" (>3.9-fold) and "Good healthy" (>8.9-fold) compared to the mean of the other three cultivars. Bitter gourd stems and leaves from the cultivar "Bilai" increased plasma insulin concentrations in all pigs combined by 28% (from 53±6 to 67±9 pmol/L, p<0.03). The other two cultivars of stems and leaves showed consistent numerical but no significant increases in plasma insulin concentrations. The effects on plasma insulin concentrations were confirmed in mildly D pigs but not in overtly D pigs.

## Conclusions

Fruits of Bitter gourd improve glycemic control and stems-leaves of Bitter gourd increase plasma insulin concentrations in an obese pig model for mild diabetes. The effects of Bitter gourd fruit on glycemic control seem consistent but relatively small and cultivar specific which may explain the varying results of human trials reported in the literature.

## Introduction

*Momordica charantia*, also known as Bitter melon, Sopropo, Karela or Bitter gourd is used in folk medicine all over the world as functional food for the treatment of different pathologies, mainly obesity, metabolic syndrome and type 2 diabetes mellitus [1]. All the parts of Bitter gourd are suitable as food or feed ingredient, in particular fruit, stems and leaves [2]. Fruit and leaves of Bitter gourd contain compounds like glycosides, saponins, alkaloids and triterpenes which can lower blood glucose levels but the mode of action is unclear because it consists of an undefined mixture of compounds [3]. Scientific proof for the beneficial effects of Bitter gourd on human metabolism is however not convincing [4]. Several clinical studies in diabetic obese patients have been conducted which suggest an effect on glycemic control but scientific robustness, proper standardization and quality control remain questionable [4]. Standardization of the study design and adherence to the study protocol is more easily achieved in animal studies and these studies showed in general beneficial effects of Bitter gourd on glycemic control [5]. However, the vast majority of these animal studies was conducted in small animals like rodents but large animal models have not been used [5]. Pig models are considered ideal for studies on food and metabolism because nutritional-physiology of pigs and man are similar [6]. Minipig models for the study of human obesity and diabetes are recommended for translational research as discussed in previous reviews [7, 8].

To fill the gap between rodent research and human studies, we have developed an adult streptozotocin-induced diabetic obese Göttingen Minipig model for human obese metabolic syndrome and mild type-2 diabetes based on previous experience [9]. This adult Minipig model expresses a non-growing, yet obese phenotype, and can be used to monitor the effect of functional foods on metabolism. Also feed intake can be accurately standardized to exclude any possible confounding effects of changes in voluntary feed intake on glycemic control. Part of the glycemic lowering effects of Bitter gourd may be caused by a reduction in food intake as induced by the appetite-suppressing effects of the bitter taste [10] and certain satiety-inducing fibre types [11], both hall marks of Bitter gourd fruit. In the present study we aimed for constant feed intake throughout the study, in order to expose the direct mechanistic physiologic and metabolic effects of Bitter gourd on glycemic control in the absence of changes in feed

intake. In addition, many different cultivars of Bitter gourd exist and not all cultivars may be equally effective in influencing metabolism and glycemic control in diabetic obese subjects. Therefore, the present study aimed to compare the effects of whole fruit or stems-leaves from five different Bitter gourd cultivars on metabolic control in adult diabetic obese Göttingen Minipigs, while concurrently conducting an in-depth analysis of triterpenoid compounds within these plant samples. The triterpenoid compounds were measured using a metabolomic approach that involved LC/MS-MS analysis, and the results were subjected to identification processes through systematic comparison, including fragment MS analysis, with the Dictionary of Natural Products database.

## Materials and Methods

### Animals, housing and diet

The performed research is in compliance with the ARRIVE guidelines on animal research [12]. Experimental protocols describing the management, surgical procedures, and animal care were reviewed and approved by the ASG-Wageningen Animal Care and Use Committee (Wageningen, The Netherlands). AVD license number 40100201858.

A total of 24 female Göttingen Minipigs were purchased from Ellegaard, Dalmose, Denmark. Female pigs were chosen because they are more sensitive to metabolic abnormalities than males [13]. Multiparous, non-pregnant sows were used to reflect adult women. Average age and body weight at the start of the experiment was 2–3 years and 30–40 kg. Pigs were group housed (3 to 4 per pen, pen size 6 m$^2$) on straw bedding and were provided with three different toys to play with. The ambient room temperature was 20˚C. All pigs were adapted to the light-dark cycle–lights being on from 06:00 to 22:00 h–and feeding was provided from 15:00 to 16:00. Pigs were fed individually 1 meal of 500 g per day of a mild obesogenic diet (Table 1).

The first week, the sows received 500 g of diet, the second week 1 kg of diet, the third week 1.5 kg of diet and thereafter 2 kg of diet per meal per day. Within five months body weight was increased more than two-fold. At this obese phenotype, meal size was reduced weekly until reaching 500 g of diet per day. Empirically it was found that at a feed intake of 500 g per day, body weight of the obese sows (~85 kg) was stable over time. Energy intake was 2262 kcal

**Table 1. Ingredients of mild obesogenic diet.**

|  | Mild obesogenic diet |
|---|---:|
|  | Ingredient % |
| Barley | 10 |
| Wheat | 8 |
| Soja hulls (crude fibre 32–36%) | 36 |
| Potato protein | 4 |
| Wheat gluten protein | 5 |
| Sucrose | 20 |
| Lard | 13 |
| Cholesterol | 0.5 |
| Limestone CaCO$_3$ (powder) | 1.4 |
| Mono-Calcium phosphate | 1.8 |
| NaCl | 0.6 |
| Premix 2 g/kg | 0.2 |
| L-Tryptophan | 0.01 |
| **Total** | **100.00** |

**Table 2. Calculated nutrient composition of the mild obesogenic diet (per 500 g) and supplements per 20 g (grass, Bitter gourd fruit and Bitter gourd stems and leaves).**

| | | Mild obesogenic diet (500 grams) | Dried grass (20 grams) | Dried Bitter gourd fruit (20 grams) | Dried Bitter gourd stems and leaves (20 grams) |
|---|---|---|---|---|---|
| Dry matter | g | 463 | 18.6 | 19.3 | 18.1 |
| Crude ash | g | 29 | 3.4 | 2.0 | 3.5 |
| Crude protein | g | 65 | 3.1 | 4.4 | 2.9 |
| Crude fat | g | 69 | 0.6 | 1.9 | 0.5 |
| Crude fibre including non-starch polysaccharides | g | 137 | 8.7 | 10.2 | 11.1 |
| Sugars | g | 111 | 2.5 | 0.3 | 0.14 |
| Sugars and starches | g | 163 | 2.8 | 0.8 | 0.14 |
| Total carbohydrates | g | 300 | 11.5 | 11.0 | 11.2 |
| Gross energy | MJ | 9.50 | 0.30 | 0.37 | 0.29 |

(= 9.5 MJ) per day (Table 2) similar to the recommended dietary energy intake for humans (2000–2500 kcal per day). Feed consumption was registered daily by weighing the provided meal and weighing feed refusals. Pigs were weighed once per three weeks.

## Fruit or stems and leaves from Bitter gourd

Four cultivars of Bitter gourd (Wild-type, HTM 242, Palee and Good healthy) were cultivated outdoors and obtained from Nong Lam University, Ho Chi Minh City, Vietnam. One cultivar of Bitter gourd (Bilai) was cultivated in a green house and obtained from Fresh farma, Bleiswijk, The Netherlands. Fruit and/or stems and leaves of these five cultivars were prepared in a dried powdered form. The drying-powdering process for all cultivars was as follows: fresh clean material was cut into pieces (thick: < 5 mm) and placed in an air-vented oven at 60˚C for three days until the weight of the material remained constant. The material was turned twice daily to improve drying. The dried material was milled to powder and sealed in plastic bags. Material was stored in the dark at room temperature prior use. Dried powdered grass (Old-ambt Crop Driers, ABZ Leusden, The Netherlands) was used as isoenergetic control treatment. Dried powdered grass at an inclusion level of 4% in the pig diet is well tolerated by pigs and macronutrient composition (protein, fat and carbohydrates) of grass [14] was in the range of the composition of Bitter gourd fruit or stems and leaves. Calculated composition of the meals (per 500 g) and the supplements (Bitter gourd or grass) per 20 g is shown in Table 2.

Composition was based on data obtained from the Centraal Veevoeder Bureau 2011, CVB table pigs, Product Board Animal Feed, The Hague, The Netherlands [14]; additional information for dried grass: Oldambt Crop Driers, harvest 2018, ABZ Leusden, The Netherlands; for Bitter gourd fruit: Agro Control, Stichting control in food & flowers, Delftgauw, The Netherlands; for Bitter gourd stems and leaves: Schothorst Feed Research, Lelystad, The Netherlands and from references [15–17]. Analyses of residues in dried powdered Bitter gourd by mass spectrometry showed no pesticides or heavy metals above legally-tolerated concentrations. Acetamiprid Q, Azoxystrobine Q, Imidacioprid Q and Cypermethrin Q were below 0.04, 0.05, 0.08 and 0.45 mg/kg and Arsene Q, Cadmium Q, Mercury (hg) Q and Led (Pb) Q were below 0.05, 0.01, 0.01 and 0.03 mg/kg (Agro Control, Stichting control in food & flowers, Delftgauw, The Netherlands).

## Induction of diabetes by streptozotocin treatment

Obese Göttingen Minipigs (~ 85 kg) were anesthetized with intramuscular azaperone 2 mg/kg (Stressnil, Janssen, Tilburg, The Netherlands), followed by intravenous thiopental 15 mg/kg

(Nesdonal, Rhone Merieux, Lyon, France). A permanent blood vessel catheter (Becton Dickinson, Secalon Seldy, 16 G, polyurethane, Franklin Lakes, NJ, USA) was inserted in the ear vein and fixed firmly to the ear. The catheter was flushed with physiological saline and sealed off with physiological saline containing 5 IU heparin per mL when not in use.

A diabetic state in the pigs was induced by slow injection (over a period of 1 minute) of the pancreatic β-cell cytotoxin streptozotocin (STZ, Enzo Life Sciences, Raamsdonksveer, The Netherlands) in the ear vein after overnight fasting, modified as described previously [9, 18, 19]. STZ was dissolved in 0.1 mol/l Na-citrate, pH 4.5 at a concentration of 1 g per 20 mL and filter-sterilized before use. STZ-injected Minipigs were provided with an afternoon meal to counteract possible hypoglycemia which can occur due to endogenous insulin release by destroyed pancreatic β-cells. Multiple daily injections of STZ are needed to induce post-STZ hyperglycemia (>10 mmol/L) and the number of STZ injections are pig specific (due to inter-animal variation). We chose to use different strategies (based on dose and number of STZ injections) to induce mild or more severe diabetes in pigs:

Strategy 1 (aim to induce mild diabetes): daily repeated STZ (1 g/day) injections were given to 10 Minipigs until reaching a >3-fold increase in fasting blood glucose concentrations. Mean baseline 18-h fasting blood glucose concentrations were 2.8±0.2 mmol/L. Three to 6 injections of STZ per pig were needed to reach a mean fasting blood glucose concentration of 11.4±0.7 mmol/L on the day after the last STZ injection. At 3, 5 and 8 weeks post-STZ treatment, fasting blood glucose concentrations slowly stabilized at 8.5±1.3, 5.9±0.7 and 4.2±0.4 mmol/L, respectively. Twelve weeks post-STZ treatment, at the start of the Bitter gourd trials, fasting plasma glucose concentrations were 4.6±0.5 mmol/L.

Strategy 2 (aim to induce overt diabetes): two injections of STZ (1.5 g on the first day and 3 g on the second day) were given to 9 Minipigs. Mean baseline 18-h fasting blood glucose concentrations were 2.9±0.1 mmol/L. Two weeks post-STZ treatment fasting blood glucose concentrations were 14.1±1.1 mmol/L. However, 4 out of 9 Minipigs showed persistent reductions in feed intake and were excluded from the study. Eight weeks post-STZ treatment, at the start of the Bitter gourd trials, fasting plasma glucose concentrations were 9.8±2.2 mmol/L.

Strategy 3 (aim to induce overt diabetes): two injections of STZ (1 g on the first day and 2 g on the second day) were given to 4 Minipigs. These 4 Minipigs were added to the study from trial 5 onwards. Mean baseline 18-h fasting blood glucose concentrations were 2.5±0.2 mmol/L. Three weeks post-STZ treatment fasting blood glucose concentrations were 14.4±3.1 mmol/L. No reductions in feed intake were observed. Eight weeks post-STZ treatment, at the start of the Bitter gourd trials, fasting plasma glucose concentrations were 9.6±1.8 mmol/L.

One obese Minipig was used as non-STZ treated reference.

## In vivo testing of Bitter gourd and timeline

Two to 3 months after the induction of diabetes, pigs were used to investigate the metabolic effects of Bitter gourd. First, the pigs were habituated to the taste of Bitter gourd (fruit or stems-leaves) and of grass over a period of three weeks. Thereafter testing of the various cultivars of Bitter gourd was started. The cross-over design per trial is represented schematically in Fig 1. At the start of each period, the daily obesogenic meals (500 g) were supplemented with stepwise increasing doses of 5, 10, 15 and 20 g of test materials. To warrant complete uptake of the obesogenic diet and the supplement, the obesogenic pellets were moisturized with 100 mL water prior administration of the dry supplements to promote sticking of the supplement to the pellets.

Ten trials were conducted, each trial lasting six weeks, in consecutive order:

Trial 1: Bitter gourd fruit (cultivar "Wild-type") versus grass

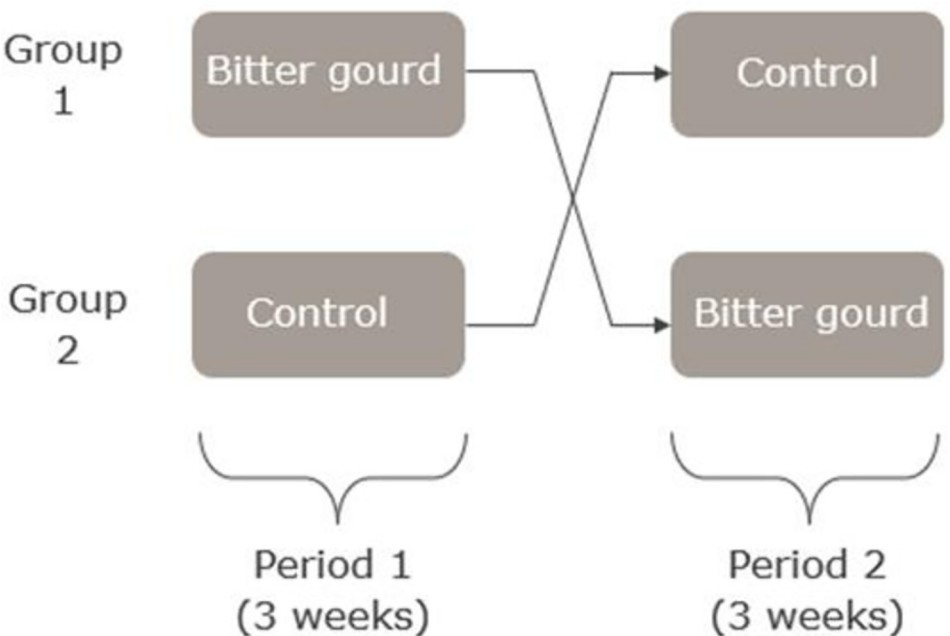

**Fig 1. Cross-over design.** Half of the pigs (group 1) received Bitter gourd during the first period of 3 weeks; the other half of the pigs (group 2) received grass (control) first.

Trial 2: Bitter gourd fruit (cultivar "HMT 242") versus grass

Trial 3: Bitter gourd fruit (cultivar "Palee") versus grass

Trial 4: Bitter gourd fruit (cultivar "Good healthy") versus grass

Trial 5: Bitter gourd stems and leaves (cultivar "Wild-type") versus grass

Trial 6: Bitter gourd stems and leaves (cultivar "Bilai") versus grass

Trial 7: Bitter gourd fruit (cultivar "Bilai") versus grass

Trial 8: Bitter gourd fruit (mix of trials 1,2,3,4,7 for average Bitter gourd effect) versus no supplement

Trial 9: Bitter Gourd stems and leaves (cultivar "Palee") versus no supplement

Trial 10: Metformin (3 g per day, maximum human dose) versus no supplement. Metformin is the most prescribed drug for the treatment of type 2 diabetes [20].

Each individual pig was followed-up per 3-week period comprising the following measurements and techniques: 1) daily food intake, 2) 3-weekly body weight, 3) at week 2, an 18-h fasting blood sample (droplet of blood) by ear vein puncture for the measurement of blood glucose, 4) at week 3, an 18-h fasting blood sample (20 mL) by puncture of the jugular vein for measurement of various parameters.

## Blood and plasma analyses

A droplet of blood was obtained from the ear vein by puncture and was analysed for glucose concentration on a blood glucose meter (On Call Extra, ACON Laboratories, San Diego, CA, USA). Larger blood samples were obtained from the jugular vein after transient sedation of the Minipigs, quickly induced by inhalation of 4% Sevoflurane (Abbott) combined with 40% oxygen and nitrous oxide via a nose-mouth cap. Both 10 mL of EDTA and 10 mL of heparin blood were collected (BD vacutainer systems, Plymouth, UK). In fresh heparin blood, blood glucose, ketones, total cholesterol, HDL, LDL and triglyceride concentrations were measured

on a blood glucose meter (On Call Extra, ACON Laboratories), on a blood ketone meter (On Call Extra, ACON Laboratories) and on a Mission Cholesterol Meter (ACON Laboratories). The EDTA and remaining heparin blood samples were centrifuged at 4000 rpm for 10 minutes at 4°C in a Rotina 35R, typ 1710 (Hettich Centrifugen, D78532 Tuttingeu) and 1 mL aliquots of plasma were stored at -80°C. Plasma insulin concentrations were measured on a porcine insulin ELISA kit (Mercodia, Uppsala, Sweden). Plasma fructosamine concentrations were measured using a nitroblue tetrazolium colometric assay (fructosamine glycated products, Abcam, Cambridge, UK). Plasma total protein concentrations were measured using the colometric Bradford assay (Abcam, Cambridge, UK). All measurements were performed in duplicate.

## Chemical profiling of Bitter gourd fruits based on LC-MS, GC-MS and HPTLC

The dried powdered samples of Bitter gourd fruit from the cultivars Wild-type, HTM 242, Palee, Good healthy and Bilai (trials 1,2,3,4 and 7) were studied for the chemical composition based on several extracts and technologies: Gas Chromatography-mass (GC-MS) spectrometry analysis, high performance thin layer chromatography (HPTLC) and Liquid chromatography–mass spectrometry (LC-MS) for targeted analysis.

The extracts for the GC/MS analysis were prepared using MeOH and n-hexane. For MeOH: Extract 30 mg of powder (15 min ultrasonication) with 1 mL of MeOH and aliquot by 100 μL. Remove the solvent with speed vacuum, take one of the aliquot and add 100 μL of pyridine containing 1 mg/mL of methyl palmitate, Add 100 μL of BSTFA with 1% TMCS and react for 60 min at 80°C in a heating block. For n-hexane: 30 mg of dried powder in 1.0 mL of hexane containing 0.5 mg of methyl palmitate, (15 min ultrasonication) and Centrifugation (10 min), Take 200 μL from the supernatant for the analysis. The extract for HPTLC was prepared as follows: 30 mg of dried powder in 1.0 mL of acetone:n-hexane (1:1) extraction and n-hexane extraction and centrifugation.

The samples were analyzed by a 7890A gas chromatograph equipped with a 7693 automatic sampler and a 5975C single quadrupole detector (Agilent, Folsom, CA, USA). Samples were separated on a DB-5 GC column (30 m x 0.25 mm, 0.25 μm film, J&W Science, Folsom, CA, USA) and eluted with He (99.9% purity) as a carrier gas at a flow rate of 1.5 mL.min$^{-1}$. The oven temperature was programmed as follows: after an initial hold at 60°C for 1 min, temperature was increased at 7°C/min to 290°C and held for 5 min, then increased to 310°C at 5°C /min and held for a further 3 min. The injector was set at 280°C and 1 μL of the sample was injected in splitless mode. The interface temperature was 280°C, and the ion source and quadrupole temperature were 230°C and 150°C, respectively. The ionization energy in EI mode was 70 eV. Compounds were identified by comparison of their retention times and ion spectra with those of the pure compounds. Data was processed using Mass Hunter (B.07, Agilent), AMDIS (V. 2.63, Agilent), and MS search (V. 2.0, Agilent). Compounds were identified using NIST MS library (version 2008).

The metabolites of Bitter gourd were further analysed by high performance thin layer chromatography (HPTLC). HPTLC chromatographic separation was performed on 20 x 10 cm HPTLC silica gel $F_{254}$ plates (Merck, Darmstadt, Germany) and samples were applied using an automatic Thin Layer Chromatography (TLC) sampler (CAMAG, Muttenz, Switzerland). Sixty mg of samples were extracted with 1.0 mL of n-hexane, and 15 μL of these solutions were applied on the TLC plate. A saturation time of 20 min was set for all chromatographic separations and the solvent migration distance spanned 85 mm from the application point. After the development with a mixture of petroleum ether–acetone–cyclohexane–ethyl acetate–ethanol

(60:10:16:10:6), the dried plates were observed under 254 nm and sprayed with NP-PEG [Natural Product Reagent (1% diphenylboryloxyethylamine in MeOH) and polyethylene glycol 4000 (5% polyethylene glycol 4000 in EtOH)]. The plate images were recorded using a TLC visualizer (CAMAG) under 366 nm UV light.

The samples were analysed using liquid chromatography mass spectrometry (LC-MS) using a UHPLC-DAD-QTOF, Thermo Scientific (Dreieich, Germany) UltiMate 3000 system coupled to a Bruker (Bremen, Germany) OTOF-Q II spectrometer with electrospray ionization (ESI). Thirty mg of dried and powdered samples were ultrasonicated for 20 min with 1 mL of 80:20 methanol:water and centrifuged for 20 min to obtain a clear supernatant (13000 rpm). The samples were filtered by 0.2 mm membrane filter. The separation was performed on a Waters $C_{18}$ column (2.1 x 100 mm, 2.1 μm). The metabolites were eluted at a flow rate of 0.3 mL/min with a gradient of 0.1% of formic acid in water (A) and 0.1% of formic acid in acetonitrile (B) of 10%-100% B in 30 minutes maintained for 5 minutes. The column temperature was assessed at 40˚C. The volume of injection was 2 μL. The mass spectrometer parameters were set as follows: nebulizer gas 2.0 bar, drying gas 10.0 mL/min, temperature 250˚C, capillary voltage 3500 V. The mass spectrometer was operated in positive mode with a scan range of 100–1650 m/z, and sodium formate was used as a calibrant.

## Statistical and data analyses

Per trial, data were analysed by the parametric two-sided Student's T-test based on paired samples (within subject, cross-over design for test-item versus control) for all obese pigs combined; for mildly diabetic obese pigs only (pigs from strategy 1) or for overtly diabetic pigs only (pigs from strategies 2 and 3). The number of pigs available per strategy group and per trial varied among trials depending on the occurrence of incomplete meal intake or any missing blood samples. Insulin resistance was estimated by a derivative of the HOMA-index [21] being blood glucose x plasma insulin and plasma fructosamine x plasma insulin concentrations. When plasma protein concentration was affected in a trial, plasma fructosamine concentration was also expressed as plasma fructosamine.protein$^{-1}$ concentration. When data were not distributed normally, data were analysed by the non-parametric Wilcoxon-test. Individual pigs were excluded from data-analysis when meals were not completely consumed throughout a trial. For Bitter Gourd fruit (trials 1,2,3,4 and 7) the data were pooled per pig to get an indication of the average effect of Bitter Gourd fruit compared to grass. For Bitter Gourd stems and leaves (trials 5 and 6) the data were pooled per pig to get an indication of the average effect of Bitter Gourd stems and leaves compared to grass. Data were expressed as Means±SEM. Effects were considered significant when p<0.05.

For chemical profiling of the dried powdered Bitter Gourd samples, the chemical composition of the samples was expressed as area under the curve per specific compound and compared between cultivars. The obtained LC-MS data was further analyzed by a supervised multivariate data analysis, principal component analysis (PCA), in which dimensions of raw data were reduced one or two principal components (PC) in order to get clustering and to reveal and visualize the systematic variation within these compound profiles.

Data files obtained from the LC–MS/MS analyses were converted to mzXML format using Brucker Daltonics DataAnalysis (version 4.1, Bremen, Germany). The LC-MS/MS data was processed using MZMine2 [22]. To build the feature matrix, mass detection was performed using centroid data. The noise level was set at 10,000 for MS and 100 for MS/MS. The chromatograms were built using the ADAP chromatogram builder [23] with a minimum number of scans of 3, group intensity threshold 1000, minimum highest intensity 10,000 and m/z tolerance of 0.05. Chromatograms were deconvoluted using a baseline cut-off algorithm with the

following parameters: minimum peak height of 10,000, peak width range of 0.02–1.00 min, and baseline level of 1000. Chromatograms were deisotoped using an isotopic peak grouper algorithm with an m/z tolerance of 0.05 and retention time tolerance of 0.1 min. The features of each sample were aligned using a join alignment algorithm with the following parameters: 0.05 m/z tolerance and 0.1 min retention time tolerance.

## Results

### Phenotype of obese Minipigs

During the 60 weeks trial period, the phenotype of mildly diabetic obese Minipigs, overtly diabetic obese Minipigs and an obese Minipig without streptozotocin treatment is shown in Table 3.

### Trials to study the effect of a test-item compared to control

Blood glucose, ketones, total cholesterol, HDL cholesterol, LDL cholesterol, triglycerides and plasma fructosamine, protein, insulin and feed intake, body weight were measured in the pigs after three week supplementation with test item compared to control.

**Trial 1: Bitter gourd fruit (cultivar "Wild-type") versus grass.** Blood cholesterol concentrations were reduced (P = 0.046) in all obese pigs combined (n = 15) by 14% (from 5.7±0.4 to 4.9±0.4 mmol/L).

**Trial 2: Bitter gourd fruit (cultivar "HMT 242") versus grass.** No significant effect for any parameter was observed in the pigs.

**Trial 3: Bitter gourd fruit (cultivar "Palee") versus grass.** Plasma fructosamine concentrations were reduced (P = 0.03) in all obese pigs combined (n = 15) by 6% (from 450±48 to 423±53 µmol/L). Changes in individual pigs are shown (Fig 2).

Plasma fructosamine concentrations were reduced (P = 0.02) in mildly diabetic obese pigs (n = 9) by 11% (from 349±26 to 309±27 µmol/L) and plasma fructosamine x insulin concentrations, a measure of insulin resistance, were decreased (P = 0.04) by 35%, (from 5201±984 to 3381±565 A.U.), i.e. insulin sensitivity was increased. Changes in individual pigs are shown (Fig 3A and 3B).

No effects were observed in the overtly diabetic obese pigs (n = **5**). Blood cholesterol concentrations were reduced (P = 0.04) in all obese pigs combined (n = 15) by 8% (from 4.6±0.3

**Table 3. Phenotype of the obese minipigs during the 60 weeks trial period, measured after 18 hours of fasting.** Blood and plasma data are calculated as the average per animal and subsequently as the mean±SEM per group. One obese minipig is shown as reference.

| | Mildly diabetic obese minipigs (n = 10) | Overtly diabetic obese minipigs (n = 9) | Reference non-diabetic obese minipig (n = 1) |
|---|---|---|---|
| Average body weight (kg) | 95 | 80 | 84 |
| Body weight gain (kg) | 1 (+1%) | -1 (-1%) | 2 (+2%) |
| Blood glucose (mmol.L⁻¹) | 3.4±0.1 | 8.5±0.3 | 2.7±0.1 |
| Plasma fructosamine (µmol.L⁻¹) | 323±12 | 583±25 | 291±12 |
| Plasma insulin (pmol.L⁻¹) | 81±4 | 40±2 | 111±4 |
| Blood total cholesterol (mmol.L⁻¹) | 4.3±0.3 | 4.4±0.2 | 3.3±0.3 |
| Blood LDL cholesterol (mmol.L⁻¹) | 1.9±0.2 | 2.0±0.2 | 0.9±0.1 |
| Blood HDL cholesterol (mmol.L⁻¹) | 2.1±0.1 | 1.9±0.1 | 2.1±0.1 |
| Blood triglycerides (mmol.L⁻¹) | 0.89±0.04 | 1.29±0.09 | 0.66±0.04 |
| Blood ketones (mmol.L⁻¹) | 0.06±0.02 | 0.16±0.09 | 0.09±0.05 |

[1] calculated (LDL = Chol–HDL–(Triglycerides / 2.2)

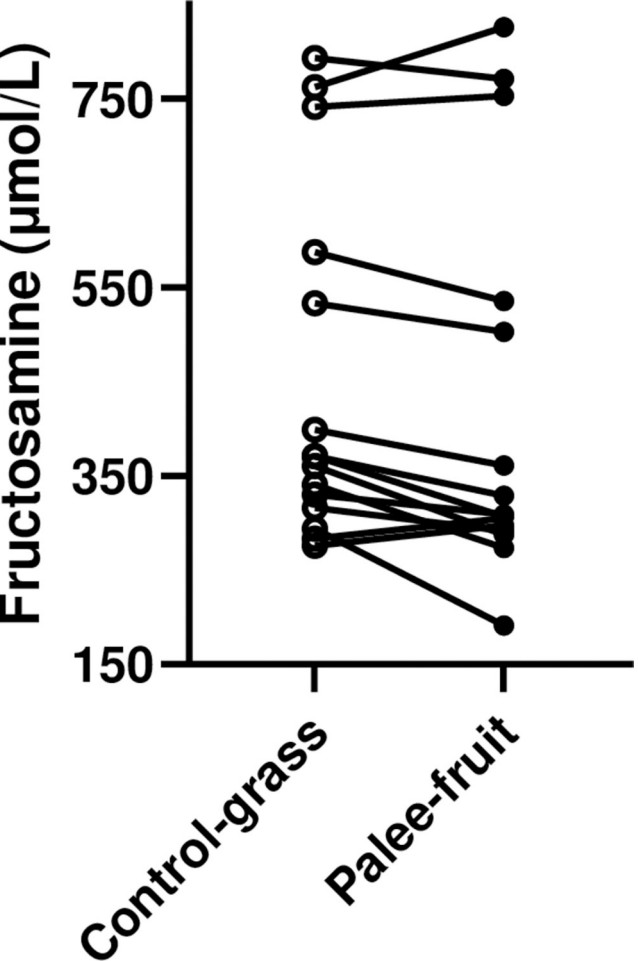

**Fig 2. Plasma fructosamine concentrations in all individual obese pigs (n = 15) fed Bitter gourd Palee fruit (black circles) or grass as control (open circles) in a paired, within-pig design.** Plasma fructosamine concentrations were reduced (P = 0.03) by Palee fruit.

to 4.2±0.3 mmol/L) and blood LDL concentrations were reduced (P = 0.01) by 15% (from 2.2 ±0.3 to 1.9±0.2 mmol/L). Body weight increased (P = 0.04) by 0.2% (from 91.8±2.7 to 92.0±2.6 kg) at identical feed intake in all obese pigs combined (n = 15). Measurements and calculations are summarized for all obese pigs combined in Table 4.

**Trial 4: Bitter gourd fruit (cultivar "Good healthy") versus grass.** Plasma fructosamine concentrations were reduced (P = 0.02) in all obese pigs combined (n = 15) by 17% (from 490 ±50 to 404±48 µmol/L). Changes in individual pigs are shown (Fig 4).

Plasma fructosamine concentrations were reduced (P = 0.04) in mildly diabetic obese pigs (n = 8) by 21% (from 384±32 to 303±22 µmol/L) as shown in the lower part of Fig 4. No effects were observed in the overtly diabetic obese pigs (n = 6).

**Trial 5: Bitter gourd stems and leaves (cultivar "Wild-type") versus grass.** No significant effect for any parameter was observed in the pigs.

**Trial 6: Bitter gourd stems and leaves (cultivar "Bilai") versus grass.** Plasma insulin concentrations were increased (P = 0.03) in all obese pigs combined (n = 20) by 28% (from 53 ±6 to 67±9 pmol/L) at increased (P = 0.04) plasma fructosamine x plasma insulin concentrations by 33%, a measure of insulin resistance (from 3152±420 to 4190±580 A.U.) and at

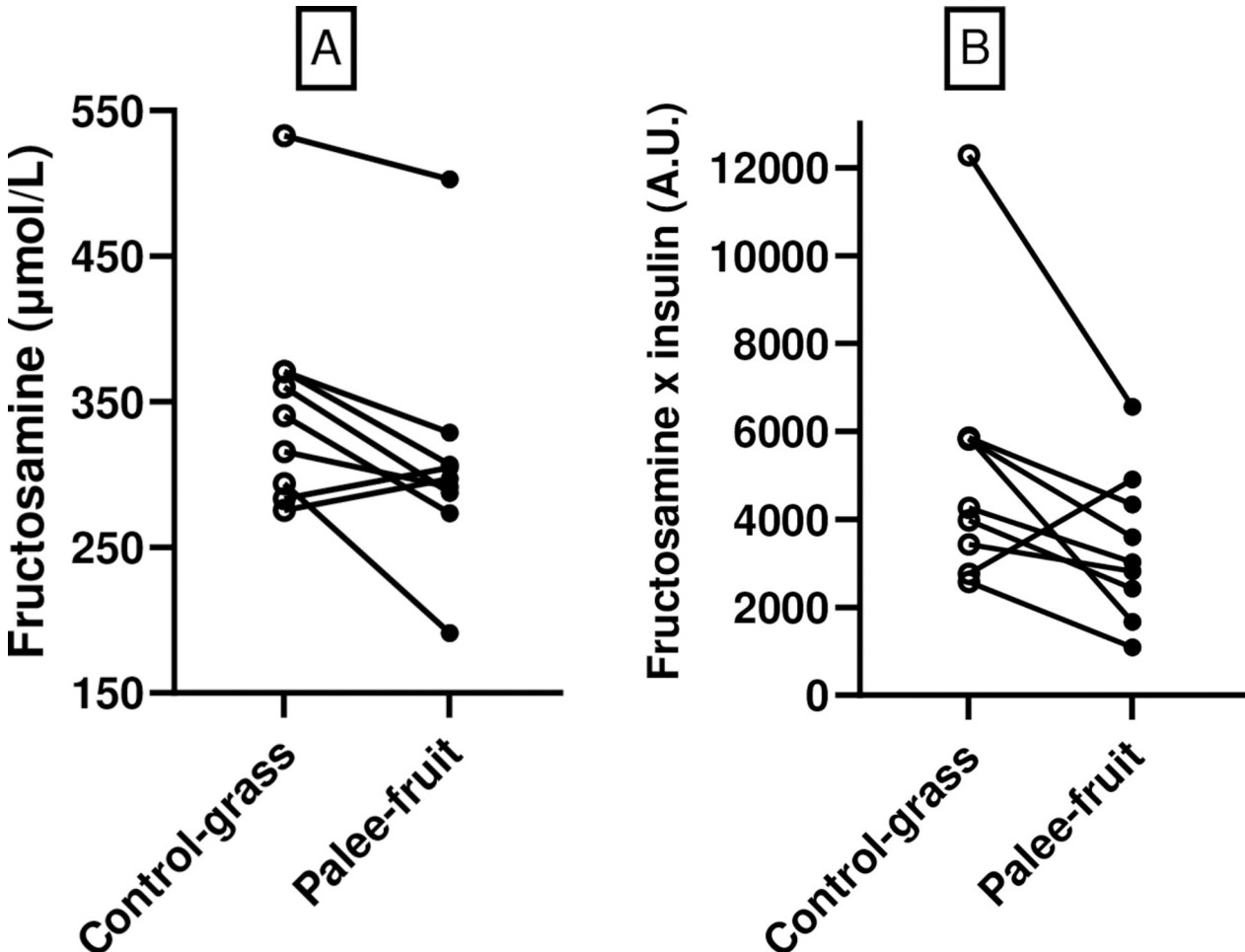

**Fig 3.** Plasma fructosamine (A) and plasma fructosamine x insulin (B) concentrations in individual mildly diabetic obese pigs (n = 9) fed Bitter gourd Palee fruit (black circles) or grass as control (open circles) in a paired, within-pig design. Plasma fructosamine and plasma fructosamine x insulin were reduced (P = 0.02 and P = 0.04, respectively) by Palee fruit.

increased (P = 0.03) blood glucose x plasma insulin concentrations by 22%, the HOMA-index of insulin resistance (from 36±4 to 44±4 A.U.). Changes in individual pigs for plasma insulin concentrations are shown (Fig 5).

Measurements and calculations are summarized for all obese pigs combined in Table 5.

Plasma fructosamine x insulin concentrations, a measure of insulin resistance, were increased (P = 0.047) in mildly diabetic obese pigs (n = 10) by 57% (from 3153±646 to 4933 ±1002 A.U.).

**Trial 7: Bitter gourd fruit (cultivar "Bilai") versus grass.** Plasma protein concentrations were increased (P = 0.049) by 5% in all obese pigs combined (n = 20) (from 96±5 to 101±6 g/ L). Plasma fructosamine.protein$^{-1}$ concentrations were reduced (P = 0.04) by 17% in mildly diabetic obese pigs (n = 10) (from 3.4±0.6 to 2.8±0.4 μmol/g). Blood cholesterol concentrations were increased (P = 0.01) by 19% in mildly diabetic obese pigs (n = 10) (from 4.9±0.4 to 5.8 ±0.5 mmol/L) and blood LDL concentrations were increased (P = 0.01) by 44% (from 2.0±0.3 to 2.9±0.4 mmol/L).

*Pooled effects of trials 1,2,3,4 and 7.* Pooled data per pig contain at least 4 out of 5 trials due to incidental incomplete meal intake or a missing blood sample. Plasma fructosamine

**Table 4. Trial 3, cultivar Palee fruit Bitter gourd tested in all obese pigs combined (n = 15).**

| | Grass mean | Grass SEM | Bitter gourd mean | Bitter gourd SEM | Delta mean | Delta SEM | Grass 100% | Grass SEM | Bitter gourd % | Bitter gourd SEM | P-value |
|---|---|---|---|---|---|---|---|---|---|---|---|
| Insulin (pmol.L$^{-1}$) | 77 | 10 | 63 | 8 | -14 | 8 | 100 | 13.4 | 82.0 | 12.3 | 0.12 |
| Fructosamine (μmol.L$^{-1}$) | 450 | 48 | 423* | 53 | -27 | 11 | 100 | 10.6 | 93.8* | 12.6 | **0.03** |
| Fructosamine x insulin(A.U.) | 5431 | 813 | 4137 | 532 | -1294 | 619 | 100 | 15.0 | 76.2 | 12.9 | 0.06 |
| Protein (g.L$^{-1}$) | 99.7 | 5.5 | 99.2 | 6.4 | -0.5 | 6.2 | 100 | 5.5 | 99.5 | 6.4 | 0.94 |
| Glucose ear (mmol.L$^{-1}$) | 5.2 | 0.9 | 5.2 | 1.0 | -0.0 | 0.3 | 100 | 17.3 | 99.4 | 19.9 | 0.94 |
| Glucose jugular (mmol.L$^{-1}$) | 4.7 | 0.7 | 4.8 | 0.8 | 0.1 | 0.2 | 100 | 16.1 | 103.4 | 17.2 | 0.33 |
| Glucose jugular x insulin (HOMA) | 50.4 | 6.5 | 45.8 | 6.3 | -4.4 | 6.4 | 100 | 12.9 | 91.0 | 13.8 | 0.49 |
| Cholesterol (mmol.L$^{-1}$) | 4.6 | 0.3 | 4.2* | 0.3 | -0.4 | 0.2 | 100 | 6.7 | 91.5* | 6.5 | **0.04** |
| HDL (mmol.L$^{-1}$) | 1.9 | 0.1 | 1.9 | 0.1 | -0.0 | 0.1 | 100 | 3.8 | 96.4 | 4.0 | 0.38 |
| LDL (mmol.L$^{-1}$) | 2.2 | 0.3 | 1.9* | 0.2 | -0.3 | 0.1 | 100 | 12.3 | 85.1* | 11.8 | **0.01** |
| Triglycerides (mmol.L$^{-1}$) | 0.95 | 0.06 | 0.95 | 0.07 | 0.00 | 0.07 | 100 | 6.6 | 100.0 | 7.7 | 0.93 |
| Ketones (mmol.L$^{-1}$) | 0.01 | 0.01 | 0.05 | 0.03 | 0.03 | 0.04 | 100 | 130 | 500 | 68 | 0.35 |
| Body weight (kg) | 91.8 | 2.7 | 92.0* | 2.6 | 0.3 | 0.1 | 100 | 2.9 | 100.2* | 2.8 | **0.04** |
| Feed intake (kg.day$^{-1}$) | 0.50 | 0.00 | 0.50 | 0.00 | 0.00 | 0.00 | 100 | 0.2 | 100 | 0.2 | 0.17 |

[1] calculated (LDL = Chol–HDL–(Triglycerides / 2.2)

*p<0.05 compared to grass. A.U., arbitrary units.

concentrations were reduced (P = 0.002) in all obese pigs combined (n = 16) by 6% (from 434 ±42 to 409±43 μmol/L). Plasma fructosamine concentrations were reduced (P = 0.013) in mildly diabetic obese pigs (n = 10) by 9% (from 354±21 to 324±19 μmol/L). Plasma fructosamine concentrations were reduced (P = 0.031) in overtly diabetic obese pigs (n = 5) by 3% (from 615±82 to 599±115 μmol/L). Changes in individual pigs for plasma fructosamine concentrations are shown in S1 Fig.

**Trial 8: Reference trial to study the effect of mixed varieties of Bitter gourd fruit compared to "no addition of grass".** Bitter gourd fruit (mix of Wild-type, HMT242, Palee, Good-healthy and Bilai, each 20%) versus no supplement revealed no significant effect for any parameter in the pigs.

Plasma fructosamine concentrations of trials 1,2,3,4,7 and 8 for mildly diabetic obese Minipigs are shown in Table 6. All trials show a numerical or significant reduction in plasma fructosamine concentrations by Bitter gourd fruit.

**Trial 9: Reference trial to study the effect of Bitter gourd stems and leaves compared to "no addition of grass".** Bitter Gourd stems and leaves (cultivar "Palee") versus no supplement revealed no significant effect for any parameter in the pigs.

Plasma insulin concentrations of trials using bitter gourd stems and leaves (trials 5,6 and 9) for mildly diabetic obese Minipigs are shown in Table 7. All trials show a numerical or significant increase in plasma insulin concentrations by Bitter gourd stems and leaves.

**Trial 10: Reference trial to study the effect of metformin compared to "no addition".** Body weight was increased (P = 0.003) in all obese pigs combined (n = 20) by 1% (from 84.0 ±3.6 to 84.8±3.6 kg) at identical feed intake. Body weight was increased (P = 0.04) in mildly diabetic obese pigs (n = 10) by 0.9% (from 93.7±1.8 to 94.5±1.8 kg) at identical feed intake. Plasma fructosamine concentrations were numerically (from 304±33 to 243±23 μmol/L, -20%) but not significantly affected in mildly diabetic obese pigs (n = 10). Changes in individual pigs for metformin are shown in S2 Fig.

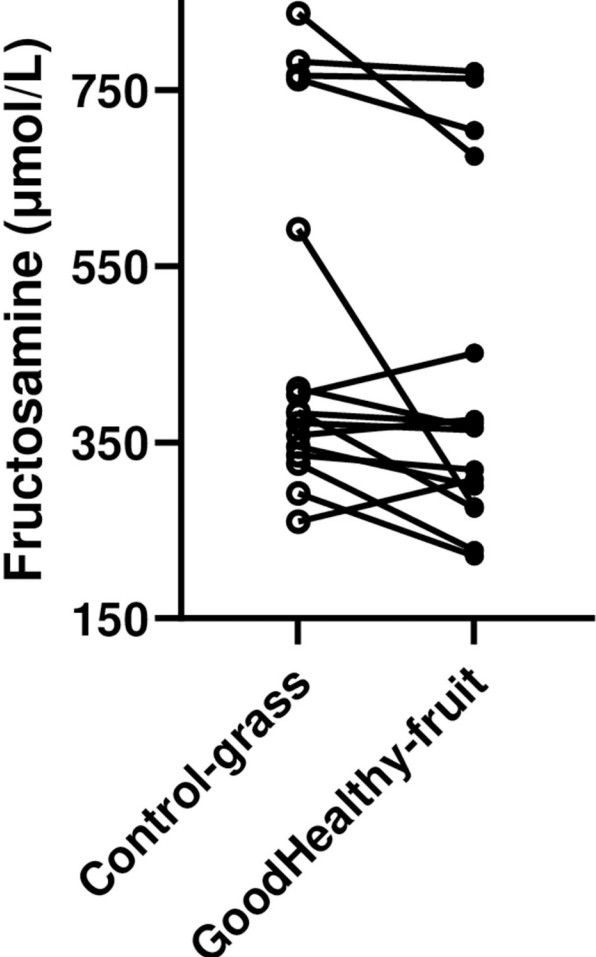

**Fig 4. Plasma fructosamine concentrations in all individual obese pigs (n = 15) fed Bitter gourd "Good healthy" fruit (black circles) or grass as control (open circles) in a paired, within-pig design.** Plasma fructosamine concentrations were reduced (P = 0.02) by Good healthy fruit.

## Chemical profiling of Bitter gourd fruits based on diverse analytical platforms

High-Performance Thin Layer Chromatography (HPTLC analysis) of Bitter gourd extracts obtained with different solvents did not reveal differences in β-carotene composition related to *in vivo* glycemic control. Also, Gas Chromatography–Mass Spectrometry (GC-MS analysis) of methanol crude extracts and of *n*-hexane extracts of the Bitter gourd samples did not result in the possibility to relate certain compounds or combination of compounds to *in vivo* glycemic control. However, our Liquid Chromatography–Mass Spectrometry (LC-MS) analysis, which was based on an 80:20 methanol:water extract of the Bitter gourd samples, yielded a more distinct chemical profile, particularly with regard to saponins. Through the quantification of compounds and subsequent Principal Component Analysis (PCA) (S3 Fig), we observed clustering among the fruit samples used for pig trials 3 and 4 (cultivars "Palee" and "Good healthy"), which displayed improved in vivo glycemic control. Interestingly, these clusters were associated with specific peaks in the LC-MS chromatograms, suggesting the presence of cucurbitane triterpenoid glycosides that were highly specific to the Bitter gourd fruit cultivars used in trials 3 and 4. These specific peaks, with retention times around 10.1 min (identified as

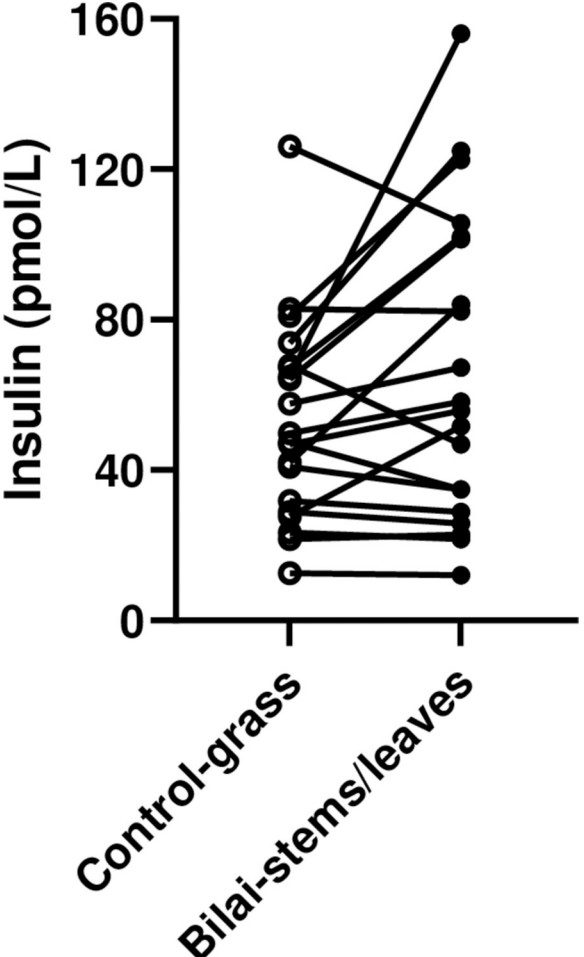

**Fig 5. Plasma insulin concentrations in all individual obese pigs (n = 20) fed Bitter gourd "Bilai" stems and leaves (black circles) or grass as control (open circles) in a paired, within-pig design.** Plasma insulin concentrations were increased (P = 0.03) by Bilai stems and leaves.

Karaviloside IX), 11.96 min (identified as Acutoside A), and 16.03 min (identified as Xuedano-side H), as shown in S4 Fig, were notably more abundant in samples from trials 3 and 4 compared to samples from trials 1, 2, and 7 (cultivars Wild-type, HMT 242, and Bilai) (Fig 6).

Calculation of the combined peak areas under the curves (AUC) for Xuedanoside H, Acutoside A, and Karaviloside IX showed that these were increased in trial 3 (the cultivar "Palee") more than 3.9-fold and in trial 4 (the cultivar "Good healthy") more than 8.9-fold compared to the mean of the other trials (trial 1, 2, and 7, the cultivars Wild-type, HMT 242, and Bilai). AUC's for Xuedanoside H, Acutoside A, and Karaviloside IX are shown in S1 Table.

These findings suggest that a combination of these specific compounds, rather than a single compound, may be associated with the observed improvement in in vivo glycemic control in trials 3 and 4. The identification of these compounds was accomplished through a rigorous process involving the comparison of their mass-to-charge ratios (m/z) and the fragmentation patterns generated by MS/MS analysis. These data were meticulously matched with established databases, including the Dictionary of Natural Compounds and data from literature [24–28].

It is important to underscore that the identification of these compounds was made with a high level of confidence, bolstered by the precise matching of their mass spectra and

**Table 5. Trial 6, cultivar Bilai stems and leaves of Bitter gourd tested in all obese pigs combined (n = 20).**

| | Grass mean | Grass SEM | Bitter gourd mean | Bitter gourd SEM | Delta mean | Delta SEM | Grass 100% | Grass SEM | Bitter gourd % | Bitter gourd SEM | P-value |
|---|---|---|---|---|---|---|---|---|---|---|---|
| Insulin (pmol.L$^{-1}$) | 53 | 6 | 67* | 9 | 14 | 6 | 100 | 12 | 128* | 14 | **0.03** |
| Fructosamine (μmol.L$^{-1}$) | 418 | 54 | 438 | 54 | 20 | 16 | 100 | 13 | 105 | 12 | 0.22 |
| Fructosamine x insulin (A.U.) | 3152 | 420 | 4190* | 580 | 1038 | 463 | 100 | 13 | 133* | 14 | **0.04** |
| Protein (g.L$^{-1}$) | 102.8 | 6.5 | 103.8 | 7.2 | 1.0 | 3.0 | 100 | 6 | 101 | 7 | 0.74 |
| Glucose ear (mmol.L$^{-1}$) | 5.7 | 0.9 | 5.7 | 1.0 | 0.0 | 0.2 | 100 | 16 | 100 | 18 | 0.96 |
| Glucose jugular (mmol.L$^{-1}$) | 5.0 | 0.7 | 5.1 | 0.7 | 0.1 | 0.3 | 100 | 15 | 101 | 15 | 0.93 |
| Glucose jugular x insulin (HOMA) | 36 | 4 | 44* | 4 | 8 | 3 | 100 | 10 | 122* | 10 | **0.03** |
| Cholesterol (mmol.L$^{-1}$) | 4.2 | 0.3 | 4.0 | 0.2 | -0.2 | 0.2 | 100 | 7 | 95 | 6 | 0.24 |
| HDL (mmol.L$^{-1}$) | 1.9 | 0.1 | 1.8 | 0.1 | -0.1 | 0.1 | 100 | 4 | 98 | 5 | 0.76 |
| LDL calculated[1] (mmol.L$^{-1}$) | 1.9 | 0.3 | 1.7 | 0.2 | -0.2 | 0.1 | 100 | 14 | 90 | 13 | 0.12 |
| Triglycerides (mmol.L$^{-1}$) | 0.89 | 0.09 | 0.95 | 0.10 | 0.06 | 0.05 | 100 | 10 | 107 | 11 | 0.25 |
| Ketones (mmol.L$^{-1}$) | 0.14 | 0.04 | 0.09 | 0.03 | -0.05 | 0.05 | 100 | 31 | 64 | 29 | 0.32 |
| Body weight (kg) | 86.8 | 3.4 | 86.7 | 3.4 | -0.1 | 0.2 | 100 | 4 | 100 | 4 | 0.55 |
| Feed intake (kg.day$^{-1}$) | 0.50 | 0.00 | 0.50 | 0.00 | 0.00 | 0.00 | 100 | 0.0 | 100 | 0.2 | 0.25 |

[1] calculated (LDL = Chol–HDL–(Triglycerides / 2.2)

*P<0.05 compared to grass. A.U., arbitrary units.

**Table 6. Plasma fructosamine concentrations (μmol.L$^{-1}$) after 18 hours of fasting in mildly diabetic obese Minipigs fed the dried powdered fruits of 5 different Bitter gourd cultivars (trials 1, 2, 3, 4, 7 and 8) compared to grass (upper panel) or compared to none (no addition) (lower panel).**

| | Grass mean | Grass SEM | Bitter gourd mean | Bitter gourd SEM | Delta mean | Delta SEM | Grass 100% | Grass SEM | Bitter gourd % | Bitter gourd SEM | P-value |
|---|---|---|---|---|---|---|---|---|---|---|---|
| **Wild-type fruit** Fructosamine | 388 | 30 | 380 | 36 | -8 | 18 | 100 | 8 | 98 | 9 | 0.68 |
| **HMT 242** Fructosamine | 370 | 27 | 366 | 22 | -5 | 20 | 100 | 7 | 99 | 6 | 0.81 |
| **Palee** Fructosamine | 349 | 26 | 309* | 27 | -40 | 14 | 100 | 7 | 89* | 9 | **0.02** |
| **Good healthy** Fructosamine | 384 | 32 | 303* | 22 | -81 | 37 | 100 | 9 | 79* | 7 | **0.04** |
| **Bilai** Fructosamine | 297 | 45 | 262 | 33 | -35 | 18 | 100 | 15 | 88 | 12 | 0.09 |
| **Pooled data from above** Fructosamine | 354 | 21 | 324* | 19 | -30 | 10 | 100 | 6 | 91* | 6 | **0.01** |
| | None mean | None SEM | Bitter gourd mean | Bitter gourd SEM | Delta mean | Delta SEM | None 100% | None SEM | Bitter gourd % | Bitter gourd SEM | P-value |
| **Mix of the 5 cultivars** Fructosamine | 339 | 46 | 326 | 51 | -13 | 23 | 100 | 14 | 96 | 16 | 0.60 |

*P<0.05 compared to grass or none.

**Table 7. Plasma insulin concentrations (pmol.L$^{-1}$) after 18 hours of fasting in mildly diabetic obese Minipigs fed the dried powdered stems and leaves of three different Bitter gourd cultivars (trials 5, 6 and 9).**

| | Grass mean | Grass SEM | Bitter gourd mean | Bitter gourd SEM | Delta mean | Delta SEM | Grass 100% | Grass SEM | Bitter gourd % | Bitter gourd SEM | P-value |
|---|---|---|---|---|---|---|---|---|---|---|---|
| **Wild-type fruit** Insulin | 61 | 12 | 82 | 10 | 21 | 11 | 100 | 19 | 133 | 12 | 0.098 |
| **Bilai** Insulin | 65 | 9 | 88 | 12 | 24 | 11 | 100 | 14 | 136 | 14 | 0.06 |
| **Pooled data from above** Insulin | 62 | 9 | 84* | 10 | 22 | 9 | 100 | 15 | 136* | 16 | **0.03** |
| | None mean | None SEM | Bitter gourd mean | Bitter gourd SEM | Delta mean | Delta SEM | None 100% | None SEM | Bitter gourd % | Bitter gourd SEM | P-value |
| **Palee** Insulin | 83 | 12 | 102 | 19 | 19 | 17 | 100 | 14 | 122 | 18 | 0.32 |

*P<0.05 compared to grass.

fragmentation patterns to established reference data. Nevertheless, it remains imperative to acknowledge that further research is essential to unveil the precise mechanisms and interactions of these compounds in relation to glycemic control. This comprehensive understanding will require additional investigations, potentially encompassing bioactivity assays, metabolic studies, and mechanistic experiments, to corroborate their roles in mediating the observed improvements in in vivo glycemic control during trials 3 and 4.

## Discussion

The diabetic obese Minipig model used in the present study is characterized by elevated fasting blood glucose concentrations, reduced fasting plasma insulin concentrations, increased plasma fructosamine concentrations and blood ketone concentrations within the normal range,

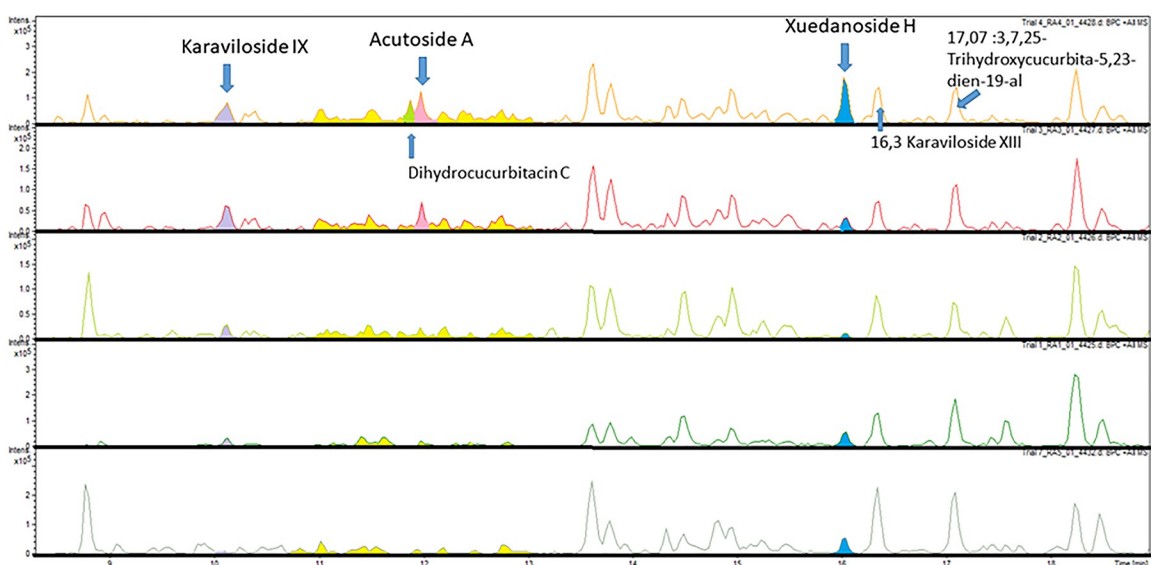

**Fig 6. LC-MS chromatograms of the cucurbitane triterpenoid glycosides from various fruit samples.** From the top to bottom, trial 4,3,2,1,7.

characteristic for obese type 2 diabetes [9]. Of note is that blood glucose concentration compared to plasma glucose concentration is approximately 50% lower in Minipigs because pig blood cells do not contain free glucose [29]. Fasting plasma glucose concentrations in the Minipigs are therefore clearly within the human range (4 to 13 mmol/L, i.e. the normal to diabetic condition) although blood and not plasma glucose was measured in the present study. With respect to dyslipidemia, the mild obesogenic diet fed to the pigs contained 20% sucrose, 13% lard and 0.5% cholesterol leading to a state of mild fasting blood hypercholesteremia, alike mild dyslipidemia in humans (LDL ~2 mmol/L) [30]. Fasting blood triglyceride concentrations in the pigs are within the normal human range (<1.9 mmol/L) [31]. As a consequence, the adult, mildly diabetic, obese Göttingen Minipig is a valuable human-sized translational model to study the mode of action of functional foods on glucose and cholesterol metabolism [8].

This translational study in diabetic obese Minipigs shows that the most effective Bitter gourd fruit cultivars "Palee" and "Good healthy" reduce plasma fructosamine concentrations in mildly diabetic obese subjects in the range of 6–21% without affecting 18-h fasting blood glucose concentrations. Plasma fructosamine concentration is a reflection of mean daily (average of fasting and prandial) blood glucose concentrations over the previous 2 to 3 week time period [32, 33]. This suggests that the beneficial effect of Bitter gourd fruit on glycemic control is mainly related to the prandial and not to the fasting period. Chemical profiling of the most effective Bitter gourd fruit cultivars (Palee and Good healthy) show that these cultivars contain higher compound intensities of the triterpenoids Xuedanoside H, Karaviloside IX and Acutoside A compared to the less effective fruit cultivars (Wild type, HMT242 and Bilai). These triterpenoids have previously been identified in Bitter gourd [24–28]. Xuedanoside H is a phenolic glycoside derivate which is known to be a SGLT2 inhibitor [34, 35]. This means that it blocks the activity of the glucose transport protein in the kidney. The result is that Xuedanoside H lowers blood glucose concentrations by increasing the flow of glucose from blood to urine. In fact, a Xuedanoside-like compound is present in the already existing anti-diabetic drug Dapagliflozin, used for the treatment of type-2 diabetic patients [34]. Karaviloside IX is a triterpene which is known to be an α-amylase and α-glucosidase inhibitor [26]. This implies that it blocks enzymes that digest starches in the small intestine. Impaired degradation of starch leads to a lower influx of glucose from the gut to blood, thereby reducing blood glucose concentrations. The anti-diabetic drug Acarbose, used for the treatment of type-2 diabetic patients, has a similar mode of action. Acutoside A is a pentacyclic triterpenoid derived from oleanolic acid with substitution by a 2-O-beta-D-glucopyranosyl-beta-D-glucopyranosyl moiety at position O-3. Pentacyclic triterpenes have antioxidant properties and can modulate diabetes by hypoglycemic bioactivity [36]. When looking at the mode of action of Xuedanoside H, Karaviloside IX and Acutoside A on *in vivo* glycemic control, it seems logic that the beneficial effects of Bitter gourd fruits Palee and Good healthy are mainly related to the prandial phase where food is digested and metabolic glucose fluxes are high. With respect to the cultivar Palee, also a decrease in plasma fructosamine x insulin concentrations was found, being a surrogate index for increased insulin sensitivity, alike the HOMA index (glucose x insulin) [21]. Therefore, the complex mixture of many bioactive components present in Bitter gourd fruit seems to have multiple modes of action to affect glucose metabolism.

When pooling the data from the five tested Bitter gourd fruit cultivars Wild-type, HMT242, Palee, Good healthy and Bilai, a general effect of Bitter gourd fruit on glycemic control is obtained. Pooling the data results in significant effects of Bitter gourd fruit on glycemic control: all obese Minipigs combined show a 6% reduction in plasma fructosamine concentrations. Bitter gourd fruit seems most effective in mildly diabetic obese pigs where a 9% reduction in plasma fructosamine concentrations is observed and in overtly diabetic obese

pigs a 3% reduction in plasma fructosamine concentrations is found. These effects seem small but in combination with changes in life style (reduction in food intake and increase in physical activity) and in combination with anti-diabetic medication, the effects may be partly cumulative. Indeed, with respect to mono-therapy using the purified anti-diabetic drug metformin, the mode of action of Bitter gourd fruit seems to be different from metformin. Metformin is known to reduce hepatic glucose production which is the primary contribution to the blood glucose-reducing effect by metformin; however, the complete mechanism of action of metformin is still not fully understood [37, 38]. By contrast, Bitter gourd fruit is known to enhance cellular glucose uptake and inhibit glucose absorption from intestine [1]. Yet, to have proof for the possible additive effects of Bitter gourd and metformin, further studies are needed.

In our mildly diabetic obese pig model, metformin resulted in a numerical but not significant reduction in plasma fructosamine concentrations by 20%. Of note, when excluding an outlier pig (which showed an unexplained increase in plasma fructosamine concentration >50% by metformin), plasma fructosamine concentrations were significantly (P = 0.021) reduced in mildly diabetic pigs by 26% (from 313±35 to 231±22 μmol/L). For reference see S2 Fig. Ineffectiveness of metformin monotherapy in the human diabetic population has been reported to be in the order of 15% [39], which is in line with the present experiment using mildly diabetic obese pigs. In any case, Bitter gourd fruit may further improve glycemic control in mildly diabetic obese subjects when combined with changes in life style and metformin monotherapy.

The small effects of pooled Bitter gourd fruit on glycemic control are illustrated by trial 8, where a mix of the five Bitter gourd cultivars was compared to "no addition" as control. In this trial a numerical (4%) but no significant reduction in plasma fructosamine concentrations by Bitter gourd was detected in mildly diabetic obese pigs. This small numerical reduction in plasma fructosamine may have been caused by the difference in nutritional load between the two pig groups. The Bitter gourd group was fed 520 g of food per day (500 g of pig pellets plus 20 g of dried powdered Bitter gourd) whereas the control group was fed 500 g of pig pellets without supplement. A difference in nutritional load of 4%. When correcting the effect of Bitter gourd by matching the nutritional load, the reduction of plasma fructosamine concentrations would be in the order of 8% (4% + 4%), reflecting the inverse correlation between meal size and glycemic control [40]. The 8% reduction in plasma fructosamine concentrations matches the outcome of the pooled data from trials 1,2,3,4 and 7 which was 9%. A difference in fibre load may also contribute to an effect on plasma fructosamine concentrations. Fibre is known to affect glucose metabolism [41]. There are many different fibre-types so it is hard to predict the net effect of fibre in Bitter gourd or in grass (as control) on glycemia.

The effect of Bitter gourd on diabetic parameters has been studied in many animal trials [1–3, 5]. Most often the study subjects were rats and mice and they all had streptozotocin- or alloxan-induced diabetes. The extracts of the different parts of Bitter gourd were orally provided to the rats or mice in the form of a powder or juice. The dosage of Bitter gourd used in the different studies ranged from 20 mg.kg$^{-1}$ BW to 1000 mg.kg$^{-1}$ BW. Most studies showed a reduction of hyperglycemia in diabetic rodents. In our pig study we used a dose of 20 g of dried powdered Bitter gourd (whole fruit or stems-leaves) per pig per day. The average pig weight was ~85 kg, therefore the dose was an equivalent of ~235 mg.kg$^{-1}$ BW. Fresh Bitter gourd fruit contains ~6% dry matter and ~94% water. A dose of 20 g of dried powdered Bitter gourd fruit corresponds therefore with ~330 g of fresh Bitter gourd fruit per day. For translational purposes to the human situation, this can be considered as a very high, almost unrealistic amount of vegetable to be consumed by humans. However, the bioactive components of Bitter gourd may also be ingested by a combination of vegetables, supplements, drinks and any other products which have been developed based on Bitter gourd. As long as the bioactive ingredients are maintained, these products might be able to add up to an active dose. The

stems and leaves of Bitter gourd contain ~40% dry matter and ~60% water. A dose of 20 g of dried powdered Bitter gourd stems-leaves corresponds therefore with 50 g of fresh Bitter gourd stems-leaves per day. This amount can be consumed by humans or can be used to make tea. Besides, the effect of Bitter gourd in pigs was found after a 3 weeks intervention while in humans one can follow diets for months which could result in a slow but steady progression towards improved markers for diabetes.

The effect of Bitter gourd fruit on blood cholesterol concentrations seems less consistent. The cultivars Wild-type and Palee reduced blood cholesterol concentrations, the cultivars HMT 242 and Good healthy showed no effect and the cultivar Bilai increased blood cholesterol concentrations. The reason for this ambiguous response is unclear but the organic bioactive compounds produced by Bitter gourd fruit are a complex blend of molecules, and the various active components could each have their mode of action on cholesterol metabolism. This underlines that the mode of action of Bitter gourd fruit seems to be cultivar specific.

Bitter gourd fruit cultivars Palee and Good healthy are good candidates to serve as a functional food for improving glycemic control in mildly diabetic obese patients but the stems and leaves do not seem to have a beneficial effect on glycemic control. Stems and leaves induce an elevation in plasma insulin concentrations in the absence of improved blood glucose or plasma fructosamine concentrations. However, stems and leaves may be interesting as supplement in livestock feed. An elevation in plasma insulin concentrations may promote whole body protein anabolism in young growing animals, thereby increasing production and reducing nitrogen excretion [42]. A recent study by us [43] showed that 0.65–1.3% inclusion of Bitter gourd stems and leaves in feed of growing pigs had no effects on plasma insulin and urea concentrations and on pig performance. These inclusion levels of Bitter gourd stems and leaves in feed were however 3-fold to 6-fold lower compared to the present study (4% inclusion level). Any possible beneficial effects of Bitter gourd stems and leaves in Livestock production require further investigation.

## Conclusions

This study shows that the Bitter gourd fruits from the Palee and Good healthy cultivar improve daily plasma glucose control in mildly diabetic obese subjects. These cultivars differed from less effective cultivars in their compound intensities of the triterpenoids Xuedanoside H, Karaviloside IX and Acutoside A. Increasing renal urinary glucose excretion (Xuedanoside H), decreasing carbohydrate digestion in the intestine (Karaviloside IX), increasing hypoglycemic bioactivity (Acutoside A) and increasing insulin sensitivity (fructosamine x insulin) might have been the mode of action. Bitter gourd stems and leaves from the Bilai cultivar increase fasting plasma insulin concentrations.

## Supporting information

**S1 Fig. Pooled plasma fructosamine concentrations in all individual obese pigs (n = 16) fed 4 or 5 different cultivars of Bitter gourd fruit (black circles) or grass as control (open circles) in a paired, within-pig design.** Plasma fructosamine concentrations were reduced (P = 0.002) by pooled fruit.
(TIF)

**S2 Fig. Plasma fructosamine concentrations in individual mildly diabetic obese pigs (n = 10) supplemented with metformin (black circles) or "no addition" as control (open circles) in a paired, within-pig design.**
(TIF)

**S3 Fig. Score plot of principal of component analysis (PC1/PC2) based on LC-MS data.**
(TIF)

**S4 Fig. The fragmentation spectra of Karaviloside IX, Acutoside A and Xuedanoside H.**
(TIF)

**S1 Table. The individual and combined peak area's under the curves (AUC in arbitrary units) for Xuedanoside H, Acutoside A and Karaviloside IX in Bitter gourd fruit from trials 1,2,3,4 and 7.**
(DOCX)

**S1 File. Individual pig data, further explanations upon request.**
(XLSX)

## Acknowledgments

We thank Hans van Diepen for formulation of the experimental diets. Piet van Wikselaar, Dirk Anjema, Bert Beukers, Jacolien van Laar, Sabine van Woudenberg, Ries Verkerk and John Jansen are acknowledged for their excellent biotechnical support. Jinke Oosterhof, Vivian van der Nat and Romy Hendricks provided both intellectual and practical assistance to the project. Rene van Rensen is thanked for obtaining dried and powdered Bitter gourd from Vietnam. Karin Senf and Steef Meewisse are acknowledged for their roles in facilitating the project on Bitter gourd.

## Author Contributions

**Conceptualization:** Sietse Jan Koopmans, Tâm Phạm Thị Minh, Nikkie van der Wielen.

**Data curation:** Gisabeth Binnendijk, Jurriaan J. Mes.

**Formal analysis:** Gisabeth Binnendijk, Allison Ledoux, Young Hae Choi.

**Funding acquisition:** Sietse Jan Koopmans.

**Investigation:** Jurriaan J. Mes, Xiaonan Guan, Tâm Phạm Thị Minh.

**Methodology:** Gisabeth Binnendijk, Allison Ledoux, Young Hae Choi, Francesc Molist, Tâm Phạm Thị Minh, Nikkie van der Wielen.

**Project administration:** Jurriaan J. Mes.

**Resources:** Tâm Phạm Thị Minh, Nikkie van der Wielen.

**Supervision:** Sietse Jan Koopmans, Francesc Molist, Nikkie van der Wielen.

**Validation:** Xiaonan Guan, Francesc Molist, Nikkie van der Wielen.

**Writing – original draft:** Sietse Jan Koopmans, Allison Ledoux, Young Hae Choi.

**Writing – review & editing:** Young Hae Choi, Jurriaan J. Mes, Xiaonan Guan, Francesc Molist, Tâm Phạm Thị Minh, Nikkie van der Wielen.

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
