## [Decision Letter · Decision Letter 0]

1 Sep 2023

PONE-D-23-02230Momordica charantia fruit reduces plasma fructosamine whereas stems and leaves increase plasma insulin in adult mildly diabetic obese Göttingen MinipigsPLOS ONE

Dear Dr. Koopmans,

Thank you for submitting your manuscript to PLOS ONE. After careful consideration, we feel that it has merit but does not fully meet PLOS ONE’s publication criteria as it currently stands. Therefore, we invite you to submit a revised version of the manuscript that addresses the points raised during the review process.

We look forward to receiving your revised manuscript.

Kind regards,

Ahmed E. Abdel Moneim

Academic Editor

PLOS ONE

Journal Requirements:

3. Please upload a new copy of Figure 6 as the detail is not clear. Please follow the link for more information: https://blogs.plos.org/plos/2019/06/looking-good-tips-for-creating-your-plos-figures-graphics/" https://blogs.plos.org/plos/2019/06/looking-good-tips-for-creating-your-plos-figures-graphics/

Additional Editor Comments:

I have additional comment as follows:

The authors need to go deeper into the study by investigating the molecular mechanisms behind their findings.

Reviewers' comments:

Reviewer's Responses to Questions

**Comments to the Author**

1. Is the manuscript technically sound, and do the data support the conclusions?

Reviewer #1: Yes

Reviewer #2: Yes

Reviewer #3: Partly

Reviewer #4: Yes

2. Has the statistical analysis been performed appropriately and rigorously? 

Reviewer #1: Yes

Reviewer #2: I Don't Know

Reviewer #3: Yes

Reviewer #4: Yes

3. Have the authors made all data underlying the findings in their manuscript fully available?

Reviewer #1: Yes

Reviewer #2: Yes

Reviewer #3: No

Reviewer #4: Yes

4. Is the manuscript presented in an intelligible fashion and written in standard English?

Reviewer #1: Yes

Reviewer #2: Yes

Reviewer #3: Yes

Reviewer #4: Yes

5. Review Comments to the Author

Reviewer #1: The manuscript represents an interesting study on the effect of Bitter gourd on a range of plasma compounds related to diabetes. The use of the diabetic pig model is highly relevant, and an impressing number of studies have been completed. The rationale for choosing mildly and overtly diabetic obese minipigs is missing and a group of obese non-diabetic minipigs would have been interesting as a control group. The chemical profiling of the Bitter gourd fruits is very interesting, however, the description of the methods is insufficient and the level of identification of the triterpenoids is not described. As these results are very interesting and the compounds are contributing to elucidating the mode of action of Bitter gourd, this should be given more focus – in the Introduction as well.

Specific comments:

Line Comment

39 How was this confirmed. This statement is unclear.

75 Which compounds? Mode of action?

Table 1 “Soja hulls RC 320-360” – is this a brand? Or what’s the reason for the name?

144 Was stems and leaves not available for all cultivars?

Table 2 Please define “Crude fiber and non-starch polysaccharides” – maybe it should be including non-starch polysaccharides?

Table 2 Analyzed nutrient composition would have been better.

242 What’s the rationale for using Metformin?

272 How were the metabolites extracted?

273 No description of the GC-MS analysis.

319-321 Very short description, a bit more details would be nice. PCA-analysis?

331 Data could be presented in a table

Table 3 Please add that it is for all pigs – and the number of pigs per treatment.

Table 4 Same as for Table 3.

455 Why against “no addition of grass”?

500 What is meant by “a more characteristic chemical pattern”? Please clarify.

502 I don’t understand the interpretation of Figure 6. In my view trial 2 is very close to trial 3 and 4.

Figure 6 could be exchanged for Supporting Figure S3, which is far more interesting.

505 What is meant by “This classification”?

514 The details on this analysis are missing in Materials and methods.

518 How were the triterpenoids identified? Authentic standards, fragmentation pattern ….? What is the level of identification?

Table 7 Move to supplementary material.

551 Has these triterpenoids previously been identified in Bitter gourd?

560 Please reformulate this sentence.

581-584 Very speculative – reformulate or provide evidence.

590-593 Why wasn’t this pig excluded?

597-598 Speculation.

605 Could the difference in dietary fiber play a role?

607-608 Can these percentages be added in this way?

619 Were there any comparable effects? Discuss the animal models.

645-648 This is very speculative.

650-664 This section could be moved to the beginning of the Discussion.

Reviewer #2: 119 Why did you choose female pigs?

Are the females pregnant or nulliparous?

This choice should be justified

Laboratory animals, especially females, are more sensitive than males and when females are pregnant or nulliparous, the tests may be influenced by these physiological characteristics. These are some of the reasons why it is important to specify the physiological characteristics of the animals tested

304 It should be remembered that diabetes is not limited to the measurement of blood sugar but also to the measurement of triglycerides which are made up of lipids and therefore form fat. As it happens, we can know that in the Krebs cycle, there is a gluconeogenesis, which form glucose from non-carbohydrate elements. The fat that is formed as a reserve is transformed and takes on the connotation of sugar ready to be used by the brain to continue feeding the body in case of prolonged fasting. In summary, a diabetic needs to control both sugar and membrane lipids and to have at his disposal foods such as bitter gourd to slow down the frequent desire of appetite in order to control the two parameters mentioned above.

I congratulate the authors for the quality and the scope of the study which reinforces the endogenous knowledge and moreover relieves the patients of this chronic pathology which gnaws rich and poor as well in Africa in Europe etc..

Reviewer #3: Here are some suggestions to improve the quality of the manuscript

Abstract

1. The methodology needs to be rephrased for clarity. How many groups? What did each group receive?

2. Did you give the same treatment to both mildly and overtly diabetic obese minipigs?

3. How many cultivars of bitter gourd did you use?

Methods

1. Line 119, what is the total number of pigs used? How many are diabetic?

2. Lines 231-242, why did you exclude cultivars HMT 242, Good health, and Palee versus grass for better gourd stems and leaves?

3. How many pigs did you use in each trial?

4. Why did you choose grass as the control diet for some trials and no supplement as a control for other trials?

5. Line 272, did you use a single extract? If no, state all the extracts used.

6. Did you perform each trial on both mildly and overtly diabetic obese minipigs?

Results

1. Lines 302-321, did you compare the insulin, cholesterol, blood glucose, and fructosamine level between the mildly and overtly diabetic obese minipigs?

2. Lines 332-352, please compare the fructosamine, insulin, cholesterol, LDL, HDL, triglycerides, and ketones between obese minipigs without streptozotocin treatment (control), mildly and overtly diabetic obese minipigs using the significant increase or decrease.

3. Indicate if there are significant changes in all the parameters evaluated between the control and treatment groups in all the trials.

4. Use symbols to indicate significant changes in tables 3-6.

5. The results for glucosamine, blood glucose, and insulin should be presented for all the cultivars.

Reviewer #4: The manuscript entitled “<momordica charantia=""> fruit reduces plasma fructosamine whereas stems and leaves increase plasma insulin in adult mildly diabetic obese Göttingen Minipigs.” has been reviewed. The authors present a very interesting work on the use of Bitter gourd and the potential use as alternative medicine for glucose and obesity control.

The animal study was complex but carefully designed and the authors have a good experience in establishing swine models for metabolic diseases, particularly the diabetic induced minipig model.

The authors have done a good job presenting the results and implications of their findings. However, several of the finds particularly to the compounds which can be related to the positive effects of the Bitter gourd should be better clarified.

Minor comments

Line 31: the term human-sized adult GM is rather confusing. Although it’s understandable what the authors would like to say, that the model of GM is representative of an adult human, human-sized minipig is not fully correct. Please rephrase. Similar for line 83.

Line 42: metabolomics” is too general. One could say “studied by metabolomics phenotying” or “metabolomics profiling”.

Line 550, Line 669: it would more advised to refer to these compounds in term of “compound intensities” rather than concentration since they were not actually quantified using chemical standards.

Major comments on section Section 492: Chemical profiling

Line 499. There is a discrepancy between what is written here and what was presented in the materials and menthods section. Line 499 mentions 80:20 methanol:water whereas in the M&M the LC-MS extractions were done using 100% methanol. Which one is correct?

Please provide more information how extracts were prepared for the LCMS analysis. The authors only mention the extraction solvent used and nothing on the sample preparation.

Line 501: PCA analysis was preformed on the data but there is no mention of how the metabolomics data was extracted, what software was used and how was the data treated before PCA analysis. Please add this information in M&M.

Line 501: Please note that the score plot of the PCA analysis shows that the Sample from trial 2 is also clustering rather close with samples from Trial 4 and Trial 3 and separating from Trial 7 and Trial 1 along the first Principal component (PC1).

Line 501: Is it correct that the authors mention quantification? Perhaps they meant compound identification. Quantification implies the use of standard curves and the use of internal standards in the analysis which is not mentioned in the materials and methods.

Line 518: How was the identification preformed? Was it done by comparison of spectra to that of metabolomics databases ? If so which databases were used?

Please provide in supplemental material the fragmentation spectra of these specific compounds.

Furthermore, the authors mention identification. To which level according to Sumner et al were identified? Level 1 using chemical standards to compare, or only annotations based on similarities to public mass spectra databases.

Please mention this, as it is important to have a better understanding of the level of accuracy in the identification of molecular features using metabolomics. Otherwise wrong conclusions can be drawn because of poor compound annotations.

Sumner LW, et al. Proposed minimum reporting standards for chemical analysis Chemical Analysis Working Group (CAWG) Metabolomics Standards Initiative (MSI). Metabolomics. 2007 Sep;3(3):211-221. doi: 10.1007/s11306-007-0082-2. PMID: 24039616; PMCID: PMC3772505.

Line 513: how was this analysis preformed more “focused”? Please clarify.</momordica>

6. PLOS authors have the option to publish the peer review history of their article (what does this mean?). If published, this will include your full peer review and any attached files.

Reviewer #1: No

Reviewer #2: **Yes: **Lamine Baba-Moussa

Reviewer #3: **Yes: **Nathan Isaac Dibal

Reviewer #4: No

---

## [Author Response · Author response to Decision Letter 0]

5 Jan 2024

Response to reviewers and the editor have been implemented in the word file: "response to reviewers".

---

## [Decision Letter · Decision Letter 1]

22 Jan 2024

Momordica charantia fruit reduces plasma fructosamine whereas stems and leaves increase plasma insulin in adult mildly diabetic obese Göttingen Minipigs

PONE-D-23-02230R1

Dear Dr. Koopmans,

We’re pleased to inform you that your manuscript has been judged scientifically suitable for publication and will be formally accepted for publication once it meets all outstanding technical requirements.

Kind regards,

Ahmed E. Abdel Moneim

Academic Editor

PLOS ONE

Additional Editor Comments (optional):

Reviewers' comments:

Reviewer's Responses to Questions

**Comments to the Author**

1. If the authors have adequately addressed your comments raised in a previous round of review and you feel that this manuscript is now acceptable for publication, you may indicate that here to bypass the “Comments to the Author” section, enter your conflict of interest statement in the “Confidential to Editor” section, and submit your "Accept" recommendation.

Reviewer #1: All comments have been addressed

Reviewer #2: All comments have been addressed

Reviewer #3: All comments have been addressed

2. Is the manuscript technically sound, and do the data support the conclusions?

Reviewer #1: Yes

Reviewer #2: Yes

Reviewer #3: Yes

3. Has the statistical analysis been performed appropriately and rigorously? 

Reviewer #1: Yes

Reviewer #2: I Don't Know

Reviewer #3: Yes

4. Have the authors made all data underlying the findings in their manuscript fully available?

Reviewer #1: Yes

Reviewer #2: Yes

Reviewer #3: Yes

5. Is the manuscript presented in an intelligible fashion and written in standard English?

Reviewer #1: Yes

Reviewer #2: Yes

Reviewer #3: Yes

6. Review Comments to the Author

Reviewer #1: (No Response)

Reviewer #2: The have well adressed my comments. The manuscrits could be accepted for publication in curent form.

Reviewer #3: The authors have revised the manuscript as suggested. However, highlighting the revision made with different colors is required for easy identification.

7. PLOS authors have the option to publish the peer review history of their article (what does this mean?). If published, this will include your full peer review and any attached files.

Reviewer #1: No

Reviewer #2: **Yes: **Baba-Moussa Lamine Saïd

Reviewer #3: **Yes: **Nathan Isaac Dibal

---

## [Editor Report · Acceptance letter]

6 Mar 2024

PONE-D-23-02230R1 

PLOS ONE

Dear Dr. Koopmans, 

I'm pleased to inform you that your manuscript has been deemed suitable for publication in PLOS ONE. Congratulations! Your manuscript is now being handed over to our production team.

Kind regards, 

on behalf of

Dr. Ahmed E. Abdel Moneim 

Academic Editor

PLOS ONE